# The temporality of uncertainty in decision-making and treatment of severe brain injury

**Mia Krogager Mathiasen**[1]*, **Lene Bastrup Jørgensen**[1,2], **Mette From**[1], **Lena Aadal**[2,3], **Hanne Pallesen**[2,3]

**1** Silkeborg Regional Hospital, Centre of Elective Surgery, RM, Silkeborg, Denmark, **2** Department of Clinical Medicine, Centre of Elective Surgery, Aarhus University, Aarhus, Denmark, **3** Hammel Neurorehabilitation Centre and University Research Clinic, RM, Aarhus University, Aarhus, Denmark

* mia_mathiasen@hotmail.com

**Data Availability Statement:** The study's minimal underlying data set (in Danish but if required, can

## Abstract

The aim of the study is to investigate how time and uncertainties of clinical action and decision-making plays out in the practical work of early neurorehabilitation in order to present new analytical ways to understand the underlying logics and dynamic social processes that take place during professional treatment of patients with severe acquired brain injury. Drawing on ethnographic fieldwork in a Danish neuro-intensive step-down unit (NISU) specialising in early neurorehabilitation, we found that negotiation of futures takes place in the modern ICU in the present by strategically building upon past experiences. We have argued that the clinical programme therefore cannot be understood only from a "here and now perspective", since the early neurorehabilitation practice is embedded in overlapping temporalities of the past, the present, and desired futures. The study discusses the underlying logics —often hidden or unnoticed—that impact clinical practice of early neurorehabilitation, in what we have termed a *logic of clinical reenactment*, a *logic of future negotiation* and a *logic of paradox*.

## Introduction

A 58-year-old man is pruning a tree in his garden. Suddenly the branch holding him snaps and the man falls to the ground, landing unfortunately on the left side of his head. From the onset of a neurologic insult (e.g. traumatic brain injury, stroke, etc.) the patient's experience of time and space changes [1–3]. Conversely, a clock begins from the health provider perspective, during which many believe an early window exists in which the brain's dynamic response to injury is heightened and rehabilitation is assumed to be particularly effective. Yet, evidence-based research has not been able to identify this "early window of enhanced neuroplasticity," therefore, the optimal time to begin rehabilitation is still not known [4]. Nevertheless, several healthcare programs on *early neurorehabilitation* are globally founded on this time-oriented notion. Early neurorehabilitation refers in general to rehabilitation interventions and mobilisation activities that begin immediately after the stabilisation of physiologic derangement in the intensive care unit (ICU), often before the patient is weaned from mechanical ventilation and vasopressor medication [5–7].

be translated to English) is added as Supporting Information files.

**Funding:** The author(s) received no specific funding for this work.

**Competing interests:** The authors have declared that no competing interests exist. The authors declare no potential conflicts of interest with respect to the research, authorship, and/or publication of this article.

This study investigates the relationship between agency and time in treatment programmes for patients with severe acquired brain injury (sABI). The aim of the study is to present new analytical ways to understand the underlying logics and dynamic social processes that take place during professional treatment of patients with sABI in a Danish intensive care unit specialising in early neurorehabilitation, called NISU [8, 9]. NISU is short for *Neuro Intensive Step-down Unit* (in Danish known as *NISA*, short for Neuro Intensivt Step-Down Afsnit). The NISU-programme consists of six hospital beds with the opportunity to open one extra bed if necessary and treats annually around 65 patients that have been referred via a regional preadmission evaluation [10]. Eighty-five percent of the NISU patients are transferred to The Early Neurorehabilitation Clinic (Early Clinic) at Hammel Neurorehabilitation Centre and University Research Clinic (HNC) once they no longer need intensive care treatment [8]. Patients admitted to NISU are all on "assessment stays," during which the professionals observe, test, and evaluate whether the patient's physical, cognitive, and/or emotional functions improve due to the treatment and early rehabilitation interventions. The study investigates how different concepts of time continuously play out in the practical work of early neurorehabilitation amongst the professional team and as such affect the clinical actions and decision-making in the treatment of patients with sABI.

## The temporality of uncertainty and decision making

To work professionally with early neurorehabilitation is a complex task given the vast diversity in the individual nature of human brain injury together with an overall presence of uncertainty towards the actual impact of clinical interventions. Two seemingly identical brain injuries often result in very different symptoms and consequences, and a rehabilitation activity that works positively for one patient might not have the same effect on the next patient [3]. Moreover, while some patients suddenly "wake up" after a severe brain injury for some unaccountable reason, other patients never regain any of their former cognitive, emotional, or functional abilities despite receiving early neurorehabilitation treatment. Due to a large proportion of so-called 'spontaneous recovery', the true impact of early interventions is difficult to assess [4].

The acute treatment of acquired brain injury has been in focus both in Denmark and internationally for many years, with good results [3, 11]. More people survive serious trauma and diseases, which has resulted in more people living with or at risk of having complex negative health effects after acquired brain injury. The objective of early neurorehabilitation besides higher survival rate is to reduce the negative effects of the disease or trauma and to enable the person to achieve an independent and meaningful life. An Danish analysis assessed the potential economic benefits of rehabilitating people with acquired brain injury [3]. However, rehabilitation as a comprehensive neurorehabilitation programme could not be compared with the option of no rehabilitation. Nevertheless, some aspects of neurorehabilitation positively influence mortality, nursing home costs and labour market participation. The health technology assessment concluded that treatment and rehabilitation for people with acquired brain injury are costly, and the early intensive part is most expensive. Neither the clinical effects nor the economic benefits of neurorehabilitation as an overall programme can be compared with the costs since the benefits cannot be quantified precisely enough [3]. Even though the Danish healthcare system seems to be set up to deal with severe brain injuries as effectively as possible, overwhelming tensions might be found in several forms: resource allocations, ethical dilemmas in relation to whether intensive early rehabilitation makes sense in proportion to outcome and matter of time–e.g. when to stop or maintain treatment of a patient.

Previous research has demonstrated how time is used as a tool by nurses and therapists to make narrative stories that guide the professionals' actions in intensive care [12, 13]. Our

study adds to this field of research by contributing with novel theoretical perspectives on how the underlying concepts of time influence not only the individual professional's plan of action, but the entire structural setup of the intensive programme of early neurorehabilitation. Moreover, the study brings forward new empirical knowledge regarding the question of the social factors that influence interdisciplinary decision-making in neurorehabilitation practices. This empirical focus has been called upon, as studies have shown how definite decision-making plays a significant role in the efficacy of interdisciplinary teams in terms of the outcome of rehabilitation [14, 15].

The clinical implications of our findings and discussions might help to raise the clinicians' awareness of their complex field of work by providing them with a new language and theoretical understanding of the various logics and temporal interplay their everyday observations, actions and decisions are made upon.

## Method

### Design

Empirical data was gathered during ethnographic fieldwork [16] in the hospital ward of NISU from January–March 2017. Qualitative methods of observation [17, 18], ethnographic interviews [19], and a focus group interviews [20, 21] were chosen to gain insight into the various social and cultural processes that influence the clinical work at NISU. The ethnographic fieldwork was conducted as an open-ended emergent learning process in which Principle Investigator (PI), an anthropologist who had never been in an ICU, over time discovered the social rules and meanings of importance within the cultural system of the intensive health care facility. Hence, the aim of the study was not decided prior to the researcher's arrival at NISU.

In the role of an *observer as participant* [17] with minimal involvement in the social setting, PI observed occupational therapists, physiotherapists, and intensive nurses and doctors in their daily interactions with ten patients with sABI during their work tasks of rehabilitation activities, caregiving, and medical treatment. She witnessed how the professionals organised mutual work strategies around the patients during morning and afternoon briefings, activity scheduling, and ward rounds. Weekly video conferences were attended, where professionals from NISU and experts from HNC discussed the recovery status of each NISU patient and exchanged knowledge on the strategy of future neuro interventions. Furthermore, PI observed three meetings with family members of admitted NISU patients, during which the professionals informed the relatives about the status of the patient and involved them in the rehabilitation programme by attaining personal information about the patient prior to the brain injury from the relatives.

The primary investigator dressed in attire similar to other healthcare workers and wore a badge identifying the researcher role. The observer role proved non-disruptive to the teams, which are accustomed to those in observer roles including students and external providers.

### The context: An intermediate hospital station

NISU is a rather unique organisation in the context of a Danish hospital. Although physically located at Silkeborg Regional Hospital, NISU is an outstation of the established neuro hospital, HNC [8, 22]. NISU is an intermediate hospital station balancing activities of *life-saving* intensive care treatment and *life-making* neurorehabilitation.

The professional team at NISU consists of neuro-specialised physiotherapists, occupational therapists, specialists in anaesthesia and intensive care, and intensive care nurses. Ancillary specialists in neurology, medicine, radiology, and laboratory services work alongside to help support the function of the patient's vital organs, management of nutrition, and to fend off

infections and other secondary consequences of the brain injury. The integration of disciplines at NISU is fully in line with the latest research showing that working in interdisciplinary teams is essential for improved outcomes for patients with brain injury [15, 23–25].

Many patients admitted to NISU have reduced respiratory function and have lost their ability to swallow and are in need of mechanical ventilation [8]. In order to secure their airways and prevent lung infections, many patients have a so-called "cuffed tracheostomy tube". A prime function of the professional team is to help the patients gradually get off the support of medical ventilation and to safely decannulate the tracheostomy tube (hence the "step down" in the name of the unit). At NISU, this weaning process is seen as a significant part of rehabilitation training and is, therefore, primarily handled by the occupational therapists and physiotherapists, although the weaning process happens in close daily cooperation and dialogue with the interdisciplinary team of nurses and specialists. However, the anaesthesiologists always have the final say and overall responsibility for the decannulation process.

**Data collection.**   The fieldwork was based on 130 hours of observations where informal, handwritten interviews with nurses, development nurse, therapists and doctors were part of the observational fieldwork. In addition, three formal (i.e. recorded) semi-structured interviews were conducted with physiotherapist, a course participant (intensive care nurse) and a spouse of a patient, and a focus group interview with two doctors, two nurses, two occupational therapists, and one physiotherapist using open-ended questions raised during the observations from the field work. The focus group interview was chosen as method in order to create a common venue from where the different professional groups would be able to elaborate and discuss subjects in regard of the uncertainties of decision-making at NISU. The focus group interview lasted about 1.5 hours and included three main topics of discussion: 1) express how you understand your particular role or function as a professional as part of the NISU team, 2) tell about a patient treatment course that you found particular challenging in respect to uncertainty in decision-making and the reason why, 3) discuss what type of 'tools' or systems you as professionals use in order to asses and decide the next step in a patient's course of treatment and future "rehabilitation potential".

The fieldwork was carried out in the entire hospital ward of NISU, where a total of ten patients was admitted in the research period. However, in interviews with the professionals and observations, focus was in particular given to the treatment programme of three senior patients (Lars, Bjarne and Helen), for which reason the analysis primarily are based on the activity around these patients. The patients were chosen because they arrived at the outset of the research period, so the progress of treatment could be observed from the beginning.

After three months of fieldwork themes regarding clinical actions, the impact of time, uncertainty and decision-making were observed in the data. To enhance trustworthiness and credibility, the processes of data collection and generation were discussed and clarified during regular research team meetings, where PI analyses were validated by co-investigators. Furthermore, the study endeavored data triangulation (involved time, space, and persons) and investigator triangulation (involved multiple researchers: a specialised nurse, three qualitative researchers with two experienced in the field of neurorehabilitation and one familiar with anthropological theories). Moreover, theory triangulation that involved several theoretical schemes are used in the interpretation of the phenomenon [19].

**Theoretical framework.**   The analysis of the study presented in the discussion section draws on theoretical concepts of time and theories of uncertainties found in contemporary anthropology [26–28] and social geography [29]. As will become apparent throughout the paper, the logics and social processes that take place during neurorehabilitation treatment at NISU cannot be understood solely from a "here and now perspective," as more temporalities

play out simultaneously in the clinical field. The agency of the present often takes the future, e.g. dreams, hopes, or anxiety for a certain future, as its starting point [26, 30–32].

With his concept *anticipatory action* [29], social geographer Ben Anderson argues that liberal democracies are repeatedly acting in advance of the future in order to prevent or prepare for societal threats or catastrophes. Recent examples of such anticipatory action can be seen in relation to the war on terrorism, trans-species epidemics, and climate change, where "bombs are dropped, birds are tracked, and carbon is traded on the basis of what has not and may never happen: the future" [29: 777]. Although one could say that 'the catastrophe' (the brain injury) has already happened in our case, the concept of anticipatory action is still interesting for the present analysis. Firstly, if you look through the eyes of the individual NISU-patient, another potential catastrophe might lurk ahead if the health care professionals decide that nothing more can be done to help you to recover and they therefore cease rehabilitation efforts with consequences as a future life with little or no brain function, a constant need for care and supervision along with the risk of early death. Secondly, the NISU programme can be viewed as a special hands-on form of anticipatory action as the professionals' actions are fundamentally based on certain anticipations about *how the future will turn out* if they choose *not* to intervene immediately (e.g. before "the window of time" closes for the patient).

Hence, the function of the theoretical framework is to recognise the categorizations of findings, as well as to open a discussion about the impact (hidden or unnoticed) concepts of time and uncertainties have in clinical practice of early neurorehabilitation, in what we have termed *logic of clinical re-enactment*, *logic of future negotiation* and *logic of paradox*.

**Analysis.** To analyse the empirical data, the transcribed interviews and observation field notes were stored, organized, and analysed by use of NVivo 11 Qualitative data analysis software (QRS International Pty Ltd, Doncaster, Australia). The empirical data was read and re-read a number of times to conduct open coding to identify 'social patterns' that went across the reflections and statements in the interviews, the focus group interview, and the observations from the hospital ward [18, 19]. The data were coded into preliminary analytical categories and subcategories to create an understanding of the place, human and nonhuman actors and activities that shaped the social setting of NISU. Through this process, the research group found that the themes of time, uncertainties and underlying logics impact clinical practice and organisation of NISU and were relevant to explore further.

Hereafter we will delve into the empirical data in order to present the environmental social setting of NISU, then the practical work of interventions carried out in the ICU, and finally the clinical assessment practice. The empirical data will be analysed and discussed using theoretical perspectives of time and uncertainties. In the discussion section, we discuss the findings and unfold the underlying logics which became visible in the empirical data.

## Ethical considerations

Ethical approval for this study was obtained by the Biomedical Research Ethics regional committee (1-16-02-91-17) and the Danish Data Protection Agency (journal no. 2012-58-006). The study was completed in accordance with the Helsinki Declaration 2018. Surrogate informed written consent was obtained from the patients' closest relatives. Consent covered both study participation and publication. Participation was voluntary, and withdrawal was possible at any time. With the exception of the named hospital departments, all other names are anonymised in the study. Moreover, the dates of observations and interviews have been removed to maintain anonymity.

## Findings

During the fieldwork the researchers identified that the health professionals were acting in a social context of uncertainty in which they constantly had to assess and decide what kind of intervention potentially would benefit the recovery of each individual patient in order "to reach into" each patient and reach their estimated "potential for recovery". Several main and subcategories emerged during the process of analysis and shaped underlying logics (see Fig 1).

## The space of opportunities

**A walk through the hospital corridor.**   In patient room 5, a 58-year-old man is lying in a hospital bed with his eyes closed. The bed is placed in the centre of the room encircled by various technical equipment attached to a ceiling mounted pendent of two flexible, metallic arms. On a screen to the left of the patient, changing graphs of the man's vital parameters light up. Several photographs of the man and his family, taken before the accident in the garden three weeks ago, hang on a white board with magnets placed in between two large windows to the outside. At a computer desk outside, a nurse looks through the window on a regular basis to check patient. She avoids disturbing him while he rests after his rehabilitation session earlier this morning. Walking through the hospital corridor, seven similar single patient rooms appear, each with white walls, circadian lighting, family photos, and a window to the corridor. A computer desk is placed beneath each window. In the heart of the unit is the nurses' station, which at certain times of the day acts as an assembly point for staff briefings where the patients' recovery statuses are discussed. Here a timetable with the planned activities for each patient's day is written on a white board, and a large monitor screen linked to all the patients echoes with a constant piping-noise. At this time, people in blue, green, or white uniforms are occupied with different tasks around the unit. A specialist in anaesthesia and an intensive care nurse deliberate the possible reasons for one patient's increasing infection score as part of the daily ward round. In the patient room opposite, a nurse injects liquid medicine though a plastic tube into an elderly male patient. Meanwhile, a physiotherapist and another nurse walk past with a female patient, who lies on a portable bath bed covered in white towels. They

| Main categories - of actions and dynamic social processes in the clinical environment | Sub categories - of actions and dynamic social processes in the clinical environment in which underlying logics appeared | Underlying logics - of clinical reasoning and decision-making |
|---|---|---|
| **The space of opportunities** **A walk through the hospital corridor** **The clinical work** **A step into a patient's first week of admission** **The assessment of potential** **A future becomes visible** | The clinical interdisciplinary collaboration Tests and measurement of progress The notions of activity and strategic use of embodied memory Anchored in uncertainty Scoring tools - a type of navigation system The essential tools for decision making The right call | **A logic of clinical reenactment** **A logic of future negotiation** **A logic based in paradox** |

**Fig 1. Overview of findings and discussions.**

disappear inside a bathroom facility with a special technical system that makes it possible to connect a mechanical ventilator directly to the bathroom wall.

The ethnographic description above portrays a typical morning in the hospital ward of NISU and reveals a place where quietude and business actions both prevail. People in different roles and life circumstances interact in a particular social environment that has specially been constructed to help people with sABI and intensive care needs to recover. With the hospital unit's single patient rooms, special lightning, physical space to move and position the patients during training sessions, and space for rest afterwards, in addition to tailored bathroom and outdoor facilities, NISU provides what is termed an "enriched environment" [33, 34]. As a physiotherapist explains in the focus group interview, the scene of NISU offers professionals opportunities "to act and do something":

> In these particular surroundings, we can try countless different things while we secure the patients' blood pressure. We can try to get the patients to stand up, in which case we have the staff [medical] a bit closer by, and if it does not work, we just bring the patient back down again. The possibilities are offered here [at NISU] so that we can easily try things out and there is goodwill for us to do so as well.

Being a designed space of opportunities is, in the mind of the health professionals, exactly what makes NISU special in the context of a Danish hospital, as the intensive programme attempts to create an environment with matching rehabilitation structures normally available only at non-intensive, highly specialised neuro hospitals, such as HNC. With its equivalent therapeutic setup, NISU, from the perspectives of the professionals creates the best possible conditions for intensive patients with severe brain injury to "wake up" and "find their way back," so that they in time will have a pleasant future life without constant dependence on the hospital and nursing system.

## The clinical work

**A step into a patient's first week of admission.** *Day 1.* At 1 pm, a team of porters enter NISU pushing a stretcher with a 58-year-old male patient on it. A nurse shows them to patient room 5. The male patient is named Lars. The professionals move the patient from the stretcher to the prepared hospital bed, where he is disconnected from the portable respirator and connected to the respirator in the room. The doctor arrives. She shines a light into both of the patient's eyes, checks the dressing around his injured head, and asks him to squeeze her hand. She scores the patient 3 on the Glasgow Coma Scale, which is the lowest score possible, indicating a severe brain injury.

*Day 2.* At the morning staff briefing a nurse says, "Lars was only scored 3 yesterday but he is making grimaces now". An hour later, the nurse has given the patient his medicine, washed him with a wet tissue, brushed his teeth, cleaned his tracheal tube, and changed his ostomy, then the physiotherapist arrives in patient room 5. The therapist looks at the patient and says in a loud, clear voice, "Good day, Lars. Good day, Mr. My name is Kirsten. I am a physiotherapist". The patient's eyes are closed. The therapist opens his eyelids with her fingers and shines a light into his eyes. She then takes his hand in hers and says, "Give my hand a squeeze, Lars. Yes, thank you. Do it again so I am sure. Good, Lars!" An occupational therapist enters the room. The physiotherapist tells her that Lars is able to squeeze her hand. "Great!" the colleague replies. The therapists together move the patient's arms and legs sideways in turns. Meanwhile they are repeatedly watching his face. The physiotherapist says, "We would like to see your eyes. Try to open your eyes, Lars". His eyes remain closed. She places her hands underneath

the patient's lumbar area and gives him a few gentle pushes back and forth while she looks at his face. "Trousers on! You are going to sit up". As the two therapists start to put a pair of trousers on the patient, he opens his eyes, "How nice, Lars! Can you look at me? Can you find my eyes?" asks the physiotherapist. The patient's eyes flicker around in the room. The therapists raise the patient with a lift, weigh him in the air, and move him to a wheelchair. She tries to place the toothbrush in his right hand, but it falls to the floor. Instead, the therapist places the patient's arm on top of her own arm, so he brushes his teeth through her movements. "I do not think he has pain, but I think he can feel it. It looks like it is a little annoying to him what you are doing," says the physiotherapist observing him. The other answers, "Yes, like it is a little unusual". She walks over to a drawer and finds the patient's pair of glasses, places them on his face, and says, "Maybe this will make a difference". The phone rings in the corridor. The nurse tells the patient's spouse that she is welcome to come for a visit after lunch. "Also, it would be great if you could bring some of Lars' clothes and maybe a cologne that he usually uses and recognises. We use these things as part of the training, you see. And if you have some photographs of him and the family, please bring them too," adds the nurse.

*Day 5*. The chief physician enters patient room 5 where a physiotherapist passes on her observations, "Lars opens his eyes at large stimuli, for example at sounds but there is still no eye contact. However, there is more eye opening today than over the weekend but they close again soon after stimuli". The doctor takes the patient's hand in his and says in a loud voice, "Can you squeeze my hand, Lars?" There is a small, yet clear movement of the patient's hand. The doctor repeats his request, after which he nods to the therapist and leaves the room. The therapist rolls some duvets together and lays them close around the patient's body and finally places a support pad under his knee, so the patient sits halfway raised on a firm treatment table. The therapist fetches a dish of water and says, "We need to get you washed, Lars". She moves the patient's right hand into the water. She then gives him a facecloth in his right hand and guides his movements so that he washes his own face. The patient moves his head towards the facecloth. "Well done, Lars," the therapist responds enthusiastically. In the same manner the patient's upper body is washed, after which the therapist places a roll-on deodorant in the right hand of the patient. "Try to feel what's happening now", she says and applies deodorant under both his arm bits. She places the deodorant on top of the patient's chest and from there she puts the lid back on. "Now it is time for this one. It is something else you know. What is it? It's your own face cream," she says while taking the lid off a jar of face cream. It gives a click. "You knit your brows, Lars. Yes, can you hear that?" She smears the face cream on his face. After an hour, they leave the patient.

*Day 7*. In the meeting room a neurologist, physiotherapist, occupational therapist, and two nurses greet the wife and son of the patient Lars. The neurologist goes through the record of the brain injury. "It is still a relatively short time ago the damage happened. There are already small signs that it is slowly improving", she explains. She emphasises that she cannot comment on how far Lars will be able to come or what he eventually will be able to achieve. "You have to prepare yourself for a very long process," she concludes. The wife is aware of this reality, but she hopes deeply that they will be able to collect all the missing parts of her husband into a whole bouquet in the end. "For now, we will have to wait and see what happens," answers the neurologist.

Four main points became visible from the description above which help to identify and better comprehend the underlying logics and dynamic processes that take place during the rehabilitation treatment programme at NISU.

**The clinical interdisciplinary collaboration.** The description portrays the clinical interdisciplinary collaboration, distribution of roles, and everyday interactions that take place at NISU among the professionals. An example of these social dynamics is seen in the work of

respiratory management (days 1, 2). As described, from the moment a new patient arrives at NISU, the nurses ensure that the respirator is connected correctly (day 1) and afterwards ensure that the patient can breathe seamlessly from the respirator by keeping the tracheal tube clean (day 2) and monitoring the patient's breathing. When the therapist arrives to train the patient's ability to swallow briefly and independently by stimulating the mouth during an oral hygiene session (day 2), the nurse has prepared the patient for the rehabilitation with a cleaned tracheal tube and the prescribed medication. Likewise, interdisciplinary work is apparent when the professionals share subjective observations and reflections (days 2, 5). As each professional group sees the patients in different situations, states, and time intervals, the sharing of experiences or intuitions helps the clinical team to make a collective rehabilitation strategy. While the nurses observe and treat the patient day and night, the therapists are with the patient during one-hour training sessions and the specialist may only see the patient once a day during ward rounds (see Fig 2).

The interdisciplinary setup, however, was also observed to cause challenges among the professional groups in relation to decision-making and uncertainties regarding distribution of tasks and responsibilities between the group of therapists and nurses. Especially the group of intensive-care nurses who expressed difficulties in finding their professional position in the structural design, and in the focus group interview, describe themselves as having an "octopus-function" and "to be juggling between their specialty in intensive care and rehabilitation". A course participant, who normally works as an intensive-care nurse at a regular ICU, has a similar experience during her training program at NISU. In an interview, she tells how she feels, that part of her autonomy and decision-making are reduced as part of the interdisciplinary setup at NISU:

Relative information meeting - within 72 hours

Rehabilitation plan: within first week

Pain assessment

Occupational therapeutic intervention

Nutritional assessment

Neuro physiotherapeutic examination and treatment

Neuro pedagogical strategy (if necessary)

Interdisciplinary scorings - within 72 hours

Decannulation step-down plan

Clinical examination of dysphagia

**Fig 2. Fixed interventions at NISU.**

Something special about NISU is the fact that the therapist is very much involved. We also have therapists with us [in the ICU] but not to the same degree; They come in briefly, get the patient to stand up, and then they leave again. You can feel that they are a strong presence in the day shift here. This is both good and bad. Actually, I think it might be really, really good patient-wise. Nursing-wise, however, it is something I have to get used to–that I have to put aside a great deal of responsibility. I am used to taking care of the patient. I decide. I control things. I can feel that as a nurse, one's role is much smaller [at NISU] compared to what I am used to—much smaller. The occupational therapists are planning the decannulation here. I'm used to doing that—me and the doctor. And I think it's very strange to come here and then suddenly, it's not me doing it. Then they [the therapists] do it for me. So, nursing-wise, it's actually something I've had a really hard time getting used to. I fully understand that some of the senior nurses, who have been here before the unit became NISU, feel the same way as me.

**Tests and measurement of progress.** Secondly, the description reveals how tests and progress measurements are central elements in the clinical work at NISU. As shown, the testing of stimulus-response (e.g. "squeeze my hand") begins immediately after the arrival of the patient (day 1) and other tests are performed during the therapeutic work when attempts to get a reaction from the patient or recognition of eye contact are made by the use of loud voices, light, and bodily movements (days 2, 5). The tests are used to assess whether or not the patient's functional abilities are improving over time during the rehabilitation programme.

**The notions of activity and strategic use of embodied memory.** Thirdly, the description provides important insights into a certain rationality shared among the NISU-professionals about what actions and strategies are needed in order to give people with sABI the best chances of recovery (day 5). The professionals believe that only by means of *activity* will they potentially be able "to reach" the patient's former sense of self. This line of thinking distinguishes NISU from many other Danish ICUs where common practice is to put patients to sleep medically with the intention to help "restless patients" stay calm. In a conversation during the fieldwork, a development nurse explains the different rationalities in this way:

Those hospitals act according to the notion; "as long as the patients lie still and asleep we are able to take good care of them". The method of NISU is to identify and test different stimuli, which can help awake the patient, for example through familiar smells, sounds, bodily movements or experiences.

In contrast to passive approaches, it is the activation of the patients' senses that are believed to stimulate and trigger lost bodily and cognitive memory in patients at NISU. It is the activation of familiar bodily movements, the feeling of water on one's skin during a therapeutic bath session or fresh air on one's skin at visit to the outdoor terrace, the well-known sound of a personal face cream lid opening, or the scent of one's individual cologne that are seen as the key to possible recovery.

**Anchored in uncertainty.** The fourth perspective that is brought to light through the description concerns the presence of uncertainty that is always entrenched in the work of early neurorehabilitation. The unsynchronised treatment approaches bear witness to this issue; hence, clinical interventions of early neurorehabilitation are today based on professional try-outs, ongoing tests, deliberation, and assessments. As shown above, when an intensive patient is admitted to NISU, no clear answers about the future prospects of the individual can be provided to the patient's relatives, hence, the outcome can go in either direction at this moment in

time. According to the professional team, each brain injury is unique in its characteristics and the timeframe for a brain to recover varies from case to case. Therefore, they must wait and see how the patient reacts to the rehabilitation over time (day 7). However, in a reality anchored in such profound uncertainty and with 'the early window of brain plasticity' coming to a close, some important questions arise. How do the professionals assess whether a patient is recovering, might potentially "wake up" if given more time, or will stay forever in the current stage as a stranger to the person he or she used to be before the injury happened?

## The assessment of potential

**A future becomes visible.** During an afternoon shift, a young female nurse estimates that the 58-year patient Lars has had no immediate recovery since he arrived at NISU a week earlier. Nevertheless, 47 days later, Lars is ready to move to the Early Clinic, HNC. Similarly to Lars, around 80 percent of NISU patients are transferred to the Early Clinic [8], meaning that they are circulatory and respiratory stable. This reduced group of patients will continue to train their cognitive and functional abilities in a programme of highly specialised rehabilitation. The patients who are professionally assessed to have no further rehabilitation potential are in contrast reassigned to local municipal rehabilitation programmes, care homes, or hospices, where limited or no rehabilitation is provided, meaning that the chances of long-term recovery are drastically reduced.

Given these fateful choices for the patients in question, a crucial part of the professionals' job at NISU is to search thoroughly for further rehabilitation potential in each patient and to assess whether or not the individual has progressed due to the early interventions provided or shows spontaneous remission over time (see Fig 3).

In order to perform this difficult task, the professionals were found to use a range of tests and scoring systems [35–40]. The observation below of a so-called "patient in-score" is an example of this practice:

> An occupational therapist, physiotherapist and intensive care nurse are gathered around a computer with a programme open on the screen. In front of them are two large binders with papers with titles such as *understanding*, *oral stimulation*, *mimics*, *head control*, *tonus*

The first scoring - within 72 hours.

Scoring must happen every 4<sup>th</sup> week during admission or more frequently if needed.

Scoring at discharge - within the last 72 hours before discharge from NISU.

At hospitalisation, and at least once per month, score: FIM, EFA, RLAS, FOIS.

At discharge and transition to regional or municipal hospices score: FIM, EFA, RLAS, FOIS, GOS-E

**Fig 3. The scoring procedure at NISU.**

*adjustment*, *deliberate motor function*, and *'to stand up'*. The professional team starts to give the patient a score between one and five for the various categories, where one is the lowest and five is the highest for functional ability. "We need to take point of departure in what we see right now," says the occupational therapist. Each step comes with a written explanation to guide the assessment. The physiotherapist explains that some categories do not always fit the patient, therefore, they sometimes have to deliberate where on the score rank, they think the patient fits best. Occasionally they open the patient's online journal to check if a comment previously written in the journal can help them to give the patient the correct score. Most times, however, they simply look directly at the patient through the window to assess, "Head control? Yes, he has that." The number 3 is written in the score box. The physiotherapist explains that they always score the patients every 4 weeks during their hospitalisation, or more frequently if large changes suddenly seem to happen. As a result of this, the computer programme will be able to make a graph displaying the patient's development over time.

In the description of *The assessment of potential* three important sub-themes became visible.

**Scoring tools—A type of navigation system.**   The patient in-score was conducted 35 hours after the arrival of a male patient. Similar score and testing sessions are carried out throughout the entire period of admission. The interdisciplinary team uses the results from the scores and observations to formulate shared strategies and goals to work towards whenever interacting with the individual patient. From conversations with the head nurse and a therapist, it becomes clear how the score tools provide a type of navigation system that helps the professionals to reason and act, as the scores create more objective parameters or measurements of the patients' progress in recovery:

We score our patients when they arrive, during their admission, and before they are passed on [in the system]. We do this in order to find out whether they are advancing. The scores are also vital in terms of having a more objective parameter that can be used to say to the municipality: 'It actually works what we do here', but also to see if the patients do not accept the rehabilitation. That is what a score can show, development for the better or the worse (nurse, focus group interview).

We are quick to set a short-term goal for the patient and preferably one at an activity level. In this way, we can rapidly measure if any progress happens. We set our goals based on certain criteria, which have to be measurable, and based on an activity so we can repeat it over, and over, and over, and over, and over again. So even if the patient has a bad day, we are able to try the activity once again the next day (occupational therapist, focus group interview).

**The essential tools for decision making.**   Although the professionals are aware that the score systems are not completely objective, since subjective estimations occasionally are difficult to avoid as illustrated in the patient in-score above, the score systems are still viewed as essential and necessary factors in the search for rehabilitation potential and decision-making by the professional team. Hence, as a nurse expresses the matter, "It is the only way we can see if there are any changes [in the patient], as we are so many people involved". With the tests and scores, the professionals are able to follow the patient's development over time and use the information to help them make the final (at times ethically problematic) decision about where in the hospital or nursing system the patient ought to be referred to after NISU.

**The right call.** In the focus group interview, the experience of making the final call was reflected upon and discussed among the professionals:

It is not a decision we make from one day to the next. We go through a lot of processes and many other actions to determine if there is the slightest chance that the patient can develop on one small thing: to follow a small request, become more stable, get less pain, or something else. I really do think we are good at giving them the chance, and letting it take the time it takes. I do not think I have experienced that we have sent a patient away where we have thought afterwards: "Well, was that in fact the right decision?" (physiotherapist).

A specialist replies as follows to the statement of the physiotherapist:

In contrast to what you have said about it being rather easy [to make the final decision] when we have not seen any progress, I personally think it is very difficult to write the patients out of the unit and sometimes make some of the really tough decisions. [. . .] I can at times feel a bit: "Was that the right choice?" I sometimes get the feeling that we are talking each other towards a certain decision. [. . .] As if it first has been mentioned "now we cannot do anymore," then it may spread a bit to "now we will not do anymore". [. . .] The decision is almost taken in advance and we just need to confirm that what we do is right.

In response and in accordance with her therapist colleague, an occupational therapist recalls her experience of making the final call like this in the focus group interview:

I have never experienced that the decision has been rushed or felt like . . . I have always had my documentation to look back on. We have evaluated many times on these matters, made a relatively simple goal from the start, and if there has been some kind of progress, then we would have discovered it.

The dialogue above illustrates how the work of making the final assessment calls is a practice that is given great priority and focus. The professionals are completely aware of the profound responsibility they carry for making the right call, as they know the very different realities and chances of life the patient will face as a result of their conclusion at NISU. On the one hand, they are intent on giving each patient the optimal opportunity to show recovery. However, on the other hand, a patient with no sign of any recovery over time takes up one of the few specialised hospital beds for another patient with severe brain injury in need of a chance of regaining life. The therapist group seems to share an understanding that they can only judge and act based on what they see and based on the many ongoing examinations and tests they and their colleagues perform in the reality of uncertainty at NISU. It is, however, always the anaesthesiologists who hold the overall responsibly and the final say for procedures, which might explain why the specialist finds the final decision-making especially difficult and morally challenging.

## Discussion

Our study set to explore how uncertainty temporalities of clinical action and decision-making is played out in the practical work of early neurorehabilitation in order to present new analytical ways to understand the underlying logics and dynamic social processes that take place during professional treatment of patients with sABI. Three underlying logics appear to be important, a logic of re-enactment, a logic of future negotiation and a logic of paradox.

## A logic of clinical reenactment

We identified the range of elements important to comprehend the clinical work that is carried out in the hospital unit, namely the relationship between the clinical interdisciplinary collaboration, tests and measurement of progress, the notions of activity, and the strategic use of embodied memory that all take place in a clinical reality anchored in uncertainty. This brings us to the logic related to the relationship between agency and time in early neurorehabilitation, given that the NISU-therapists actively seem to reconstruct the past in the present by bringing in previously known senses, knowledge, and personal artefacts in the rehabilitation sessions in order for the patients to remember or "re-live what use to be" in the present moment. This practice, we argue, can be viewed as a certain form of *clinical reenactment*. As anthropologist Mads Daugbjerg [27, 41] shows in his studies of American Civil War reenactment, time travel and 'living history' performances, the use of materiality (historical artefacts, uniforms, weapons etc.) plays in particular an important role in recreating an experience from the past in the present. Although the reenactment of reminiscent in our study is not a traditional educational or entertainment activity, the use of artefacts from the past (e.g. before the accident) are as described above of great significance in the rehabilitation programme at NISU. These objects consist of personal belongings of clothing, sandals, reading glasses, a face cream, toothbrush, electric shaver, and hair products, as well as recognisable personal decoration of the room with family pictures and self-made arts and crafts made by the patient before the accident in the past. Moreover, the relatives of NISU patients are strategically being involved in the reenactment process as only they can bring this unique required knowledge about the patient's former habits, personal preferences, and objects used to the training activities (day 2). In this way, the clinical interventions at NISU cannot be understood only from a "here and now perspective", since early neurorehabilitation is always embedded in overlapping temporalities of the past, the present, and certain hoped-for futures.

## A logic of future negotiation

NISU is described by the professionals not to be ordinary structure in the Danish healthcare system. On the contrary, with the establishment of the NISU-programme, a completely new 'social landscape' was formed within the hospital system. The programme created so to speak a new *situated practice* in which alternative, new futures are given a chance to grow into being [26, 42]. This line of thought is articulated by anthropologist Laura Watts [26], who argues that 'the future' should always be understood as an adjustable matter in constant movement throughout the social life. It is therefore, she says, a mistake to talk about *one* future given in advance and waiting to occur, since there are always several possible futur*es* that can grow into being depending on how people and objects move through the world. The future, Watts claims, is open for negotiation, and how the future unfolds is determined by the social landscapes in which it is situated: "The future is not out there, as though disconnected from the past or present. It is made in ongoing, everyday practices and places. The future is always situated, particular to the places where it is made" [26: 187]. However, the relatives are crucial pieces, since only they can act as witnesses of the past in the present for the professionals in their active pursuit of finding the patient's "lost self" through the early neurorehabilitation efforts (Day 5). The idea of the future as an adjustable, negotiable matter is an interesting perspective in the study of NISU. Hence, on the one hand, the function of NISU only exists because the professionals assume that the futures of the patients are still open for negotiation. With the right environment and interventions, the professionals give credence to the prospect of provoking futures alternative to 'the one future' that they trust the patients will meet if they as professionals choose not to act; that is a future with little or no brain function and a constant

need for care and supervision along with the risk of early death. On the other hand, equally rooted in the NISU-programme is the professional concept that 'the adjustability of futures' will disappear over time as 'early window of brain plasticity' closes, and with it, also the negotiation of futures as the chances of recovery for the patient are believed to vanish. Hereby not said that the professionals do not recognise that they can never definitely know, even with subjective estimations and essential tools for decision-making, if 'the adjustability of the future' is in fact completely closed. Hence, in rare cases, patients with sABI are surprisingly found to wake up after 'the window of recovery' in theory should have closed permanently. This leads us to the next logic at NISU, namely the presence of an underlying paradoxical logic.

## A logic based in paradox

By means of the research, we came to understand how a paradoxical time process begins with each new patient case at NISU. As previously described, a comprehensive and consistent assessment process begins as soon as the patient arrives at NISU and are carried on until the patient leaves the hospital unit. In this process, the professionals use numbers made viable through the tests and score systems of the patient's progress in recovery as a benchmark for their decision-making. The underlying logics of this practice can be analysed by drawing on Anderson's theoretical concept *anticipatory action* [29], as similar logics seems to exist and rule in the political landscape of liberal democracies. According to Anderson, people anticipate and act on futures through assemblies of what he terms *styles*, *practices*, and *logics*. Styles consist of a series of statements through which 'the future' as an abstract category is disclosed and related to, and these statements condition and limit how 'the future' can be intervened on. In the case of NISU, 'the early window of plasticity' is such a statement. Practices give content to specific futures, which include acts of performing, calculating, and imagining. It is through these acts that futures are made present. Anderson writes:

> Indeterminate/uncertain futures have long been made present though the ubiquitous calculations that form a constant background to life. [. . .] The result is that specific futures are made present through the domain of number, numbers which are then visualized in forms of 'mechanical objectivity' such as tables, charts and graphs [29: 784].

In the case of NISU, an *anticipated future reality* is made present by the scores and numbers in graphs of the functional development of the patient over time. This brings us to the formation of *a logic*, which Anderson defines as a coherent way in which an intervention in the here and now on the basis of the anticipated future is legitimised, guided, and enacted [29]. An example on this from the study, is the nurse explaining how the scores are vital for two reasons: firstly, to see if the patient does or does not respond to the rehabilitation, and secondly, to obtain a more objective parameter to show the municipality that their interventional actions overall work and have positive outcomes. According to Anderson, any form of anticipatory action is characterised by "a seemingly paradoxical process whereby a future becomes cause and justification for some form of action in the here and now" [29: 778]. In a similar way we demonstrate that the present of anticipated futures also in this social context of NISU become the cause and justification of certain clinical actions, reasoning, and difficult ethical assessment calls in the present moment by the professional team. However, in contrast to Anderson's top-down approach we address empirically "how anticipated futures play out, are negotiated, experienced and altered on the ground" [28] by the group of health care professionals on a daily basis. Hence, in our case the scores are used to legitimise the methodological approach and actions to themselves as professionals, as well as to other higher instances powers in the hospital sector.

### The ethical dilemma of negotiating future lives

Finally, the dialogue in the focus group interview reveals that ongoing testing of the patient's functional abilities guides the professional team in their final assessment calls and helps them to justify their choice of actions in cases where no progress can be found in the patient over a certain period of time. However, it also became clear that the existing logic is not stronger than it at time raises ethical dilemmas for some of the professionals and makes them question whether an alternative future of "abandon patients" might have been possible to pursue after all, or whether some patients would have been better of having died peacefully in his or her sleep in stead of undergoing life extending treatment. In this way, NISU represents a place in the Danish hospital system where human futures are negotiated—a place where new futures are given a chance to grow into being. However, even with this flexibility and support come ethical dilemmas that the intensive care/rehabilitation professionals must constantly try to find solutions to by using scores and navigational tools that can provide some sort of rationale that make sense in the interdisciplinary decision-making. They must perform in a tense situation balancing subjectivity and objectivity, in which social dynamics and social process shaped by professionals' opinions (e.g. projected futures, hope, uncertainty, personal relations, love, grief, and loss) affect and alter the negotiation of future lives.

Current findings are in line with other ethnographic studies of ICUs. One study found that decision-making in critical illness involves a web of discussions regarding the potential outcomes and processes of care [43]. Another concluded that in ICUs, urgency and seniority have a part to play in shaping jurisdictional boundaries at the level of day-to-day practice [44]. Future research should pay closer attention to uncertainties, decision-making, and negotiation of future lives in order to find ways to strengthen the intensive care practices of severely brain injured patients, the help provided to the relatives and to minimize health professionals' challenges in the difficult treatment of severe brain injury.

### Strengths and limitations of the study

The concepts of transferability, credibility, dependability, and confirmability were used to evaluate the scientific trustworthiness of the study [45]. The study was conducted in Denmark, and the data was embedded in a Danish cultural setting. However, the participating patients and health professional are not tied to geographical and cultural borders and therefore transferable. To increase the credibility and confirmability of the study, triangulation methods were utilized as described in the methods section [46]. Furthermore, the authors were thorough in describing the steps in the research process, aiming for transparency leaving the reader of the article able to assess the research question, methods, and findings. The limited sampling and time of ethnographic fieldwork of the current study unfortunately reduced the transferability of the variation of complexity of different types of severe brain injury patients.

## Conclusion

Severe brain injury and uncertainty go hand in hand. The results of our study allow us to recognise the underlying logics of early neurorehabilitation, as well as the continuous questions raised by the professionals towards these same logics. Although everything possible within their professional power has been tried and extensive time has been allowed, the extremely rare chances of a spontaneous inexplicable recovery long after a brain injury nevertheless continue to exist. However, without the existence of such programmes of life-saving and life-making, most people with severe brain injury would most likely be characterised as incurable and left in a bed connected to a mechanical ventilator for the rest of their life with very limited prospects of recovery. As this study shows, the negotiation of futures take place in the present

at the intensive unit at Silkeborg Regional Hospital by strategically using the past as a launch pad for something new and potentially fruitful to grow. We have argued that the clinical programme therefore cannot be understood only from a "here and now perspective", since the early neurorehabilitation practice is embedded in overlapping temporalities of the past, the presence, and certain hoped-for futures.

## Supporting information

**S1 File.**
(DOCX)

**S2 File.**
(DOCX)

**S3 File.**
(DOCX)

**S4 File.**
(DOCX)

## Acknowledgments

We would like to thank the dedicated health care professionals who participated in the study. Thanks also to the relatives of the patients who, despite very difficult circumstances, agreed to participate in interviews. They all contributed important new knowledge to the health care discipline of early neurorehabilitation.

## Author Contributions

**Conceptualization:** Mia Krogager Mathiasen, Lene Bastrup Jørgensen, Hanne Pallesen.

**Data curation:** Mia Krogager Mathiasen.

**Formal analysis:** Mia Krogager Mathiasen.

**Funding acquisition:** Lene Bastrup Jørgensen, Hanne Pallesen.

**Investigation:** Mia Krogager Mathiasen.

**Methodology:** Mia Krogager Mathiasen.

**Project administration:** Mia Krogager Mathiasen, Hanne Pallesen.

**Supervision:** Mia Krogager Mathiasen, Lene Bastrup Jørgensen, Mette From, Lena Aadal, Hanne Pallesen.

**Validation:** Mia Krogager Mathiasen.

**Visualization:** Mia Krogager Mathiasen.

**Writing – original draft:** Mia Krogager Mathiasen.

**Writing – review & editing:** Mia Krogager Mathiasen, Lene Bastrup Jørgensen, Mette From, Lena Aadal, Hanne Pallesen.

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
