## [Decision Letter · Decision Letter 0]

24 Mar 2020

PONE-D-19-35410

Time and Negotiation of Futures in the treatment of severe brain injury: An ethnographic study of anticipatory action in early neurorehabilitation

PLOS ONE

Dear Mrs. Krogager Mathiasen,

Thank you for submitting your manuscript to PLOS ONE. After careful consideration, we feel that it has merit but does not fully meet PLOS ONE’s publication criteria as it currently stands. Therefore, we invite you to submit a revised version of the manuscript that addresses the points raised during the review process.

We would appreciate receiving your revised manuscript by May 02 2020 11:59PM. To enhance the reproducibility of your results, we recommend that if applicable you deposit your laboratory protocols in protocols.io, where a protocol can be assigned its own identifier (DOI) such that it can be cited independently in the future. For instructions see: http://journals.plos.org/plosone/s/submission-guidelines#loc-laboratory-protocols

We look forward to receiving your revised manuscript.

Kind regards,

Andrew Carl Miller

Academic Editor

PLOS ONE

Journal Requirements:

2) In your Data Availability statement, you have not specified where the minimal data set underlying the results described in your manuscript can be found. PLOS defines a study's minimal data set as the underlying data used to reach the conclusions drawn in the manuscript and any additional data required to replicate the reported study findings in their entirety. All PLOS journals require that the minimal data set be made fully available. For more information about our data policy, please see http://journals.plos.org/plosone/s/data-availability.

3) Thank you for your ethics statement : "The study was completed in accordance with the Helsinki Declaration of 2018. The regional committee for biomedical research ethics in Denmark had no objections to the study. https://doi.org/10.1080/09638288.2018.1501772. Informed written consent was obtained from the participating health professionals and the relatives of patients. All personal data were de-identified."

Please amend your current ethics statement to confirm if your named institutional review board or ethics committee specifically approved this study.

Reviewers' comments:

Reviewer's Responses to Questions

**Comments to the Author**

1. Is the manuscript technically sound, and do the data support the conclusions?

Reviewer #1: Yes

Reviewer #2: No

2. Has the statistical analysis been performed appropriately and rigorously? 

Reviewer #1: Yes

Reviewer #2: N/A

3. Have the authors made all data underlying the findings in their manuscript fully available?

Reviewer #1: Yes

Reviewer #2: No

4. Is the manuscript presented in an intelligible fashion and written in standard English?

Reviewer #1: Yes

Reviewer #2: Yes

5. Review Comments to the Author

Reviewer #1: Line 46: therefore not therefor

Line 57: I would favor defining the ICU abbreviation in line 51 rather than line 57

Please be sure that the manuscript adheres to the appropriate EQUATOR network guidelines and provide citation. Likely: O'Brien BC, Harris IB, Beckman TJ, Reed DA, Cook DA. Standards for reporting qualitative research: a synthesis of recommendations. Acad Med. 2014;89(9):1245-1251.

Please state if the study was approved by an IRB. If so, state the organization whose IRB provided approval, and list the approval number in parentheses.

Please state if informed consent was obtained and from whom. Was surrogate consent allowed?

Please state if consent covered both study participation and publication of de-identified aggregate findings.

Line 182-183: Please update software reference: NVivo qualitative data analysis software (QSR International Pty Ltd, Doncaster, Australia)

Line 199: defining NSIU should be done back up on line 58

Please streamline the methods.

Please describe criteria for deciding when no further sampling was necessary (e.g., sampling saturation).

Please describe techniques to enhance trustworthiness and credibility of data analysis (e.g., member checking, audit trail, triangulation); rationale.

Reviewer #2: The article addresses an important an important topic - underlying logics of early neurorehab - using an appropriate and under-utilised approach - ethnography. In general, the paper is clearly written, laid-out well, and provides relevant information to the reader. However, I have a number of queries which may need to be addressed in a re-framing of the paper. These are outlined below:

- The theoretical contribution of the paper is somewhat blurred. While the abstract states the paper investigates how concepts of time play out and help us understand underlying logics and dynamics of of social processes. There is also mention of 'temporal strategy games'. However the bulk of the paper focuses more broadly on the nature and objectives of early neurorehab rather than an in-depth look at the social dynamics or concepts of time. More about time in the macro sense (past, present, future) than micro temporal strategies of staff.

- Rather than concepts of time, I found the focus of the paper more on the slightly different - but also very interesting -uncertainty temporalities/alternative futures of clinical actions and decision-making in early neurorehab.

- Methods: Considering the ethnographic approach, I found the methods section lacking in a number of areas:

o How were observations ‘participant’ as opposed just observation? [see Raymond Gold's typology]

o How many hours observation?

o Needs a reflective piece on impact of who did observations?

o Was there any iteration between what you set out to observe and how data collection and analysis proceeded?

o How many ethnographic interviews were done and with who? There is a mentioned of transcribed interviews (p9) not sure how evident these are in findings

o Why focus group? How long? Discussion topics?

o Are relatives in interviews or observations? ‘Participated in three meetings with family members’ (p7) how?

o Page 9 mentions hospital practice documents – not visible in findings

o Not always clear where data in findings is from (obs, interview, FG)

o Integration of data not clear – focus group dialogue appears in abrupt manner

o Triangulation of data not evident (line 736)

o Some presumptuous language about readers understanding (line 736-739)

- Linked to the above point, I have some concern about objectivity of analysis. At times analysis feels like it is selling NISU. See lines 244, 301-312, 400-403, 455-456, 616, 723-725, 749.

- Context section on NISU is far too long. Space of Opp section talks a lot about physical space but this doesn’t really feed into key findings – what does this section tell us, other than description of environment – which is then not taken up in discussion. No real discussion of time here either i.e. how time is entangled in physical space isn’t immediately obvious. Findings and discussion focus more on the decisions making and clinical action.

- Findings structured by; ‘space of opportunities, the clinical work, the assessment of potential’ which I didn't find very informative. An opportunity was missed here for deeper investigation of very important and theoretically rich topics which are only touched on. These include: decision-making rationalities, uncertain futures, how different disciplines negotiate uncertain futures – use of scoring/tools to navigate and manage uncertainty and ‘making the final call’ (564), responsibilities of final call between anaesthesiologists v other HCPs, embodied memories, active v passive rationalities of care (lines 460-463). All of these could be the main focus of the paper (rather than temporal strategies) - as they provide a richer account of the 'social processes' behind the practical clinical work.

- Must of the key findings and discussion are from the clinical work section – could focus here for re-framing of paper.

- While relatives are ‘crucial pieces’ (757), there is little emphasis on them in the findings.

- Discussion doesn’t follow through on potential contributions – how uncertainty is managed via situated/alternative futures. Overall, this paper could be much more concise in honing in on how staff manage the negotiation of futures which underpin early neurorehab clinical work.

- The link to theories in the literature are very broad, using concepts from mobile tech, history and political economy. There is little to no mention of a range of ethnographic healthcare (including intensive care) literature (see below). Could still use ‘anticipatory action’ concept but this need to be integrated more into medical/healthcare literature.

Literature which may be of interest:

Andreas Xyrichis, Karen Lowton, Anne Marie Rafferty (2017) Accomplishing professional jurisdiction in intensive care: An ethnographic study of three units - https://doi.org/10.1016/j.socscimed.2017.03.047

Nicola Mackintosh and Jane Sandall (2015) The social practice of rescue: the safety implications of acute illness trajectories and patient categorisation in medical and maternity settings - https://doi.org/10.1111/1467-9566.12339

Elisa Giulia Liberati, Mara Gorli, Giuseppe Scaratti (2016) Invisible walls within multidisciplinary teams: Disciplinary boundaries and their effects on integrated care - https://doi.org/10.1016/j.socscimed.2015.12.002

Peter Nugus and Jeffrey Braithwaite (2010) The dynamic interaction of quality and efficiency in the emergency department: Squaring the circle? - https://doi.org/10.1016/j.socscimed.2009.11.001

Jane S. VanHeuvelen (2019) Isolation or interaction: healthcare provider experience of design change. Sociology of Health & Illness Vol. 41 No. 4 2019 ISSN 0141-9889, pp. 692–708. https://doi.org/10.1111/1467-9566.12850 [talks about role of physical space]

Papers using ethnography in intensive care settings – some of which look at health technology and time e.g. https://www.ncbi.nlm.nih.gov/pmc/articles/PMC5517684/ ; https://onlinelibrary.wiley.com/doi/abs/10.1111/nicc.12032

Suspending uncertainty in order to make clinical moves – professional rationality incorporating flow of time – could align with this: https://doi.org/10.1111/maq.12557

Decision-making in uncertainty for critical illness: https://doi.org/10.1186/s12871-016-0177-2

- Overall, I found this paper to be an interesting read of an important topic. It uses a suitable, and under-used, approach in ethnography. However, I feel the paper needs a major re-frame to cut unnecessary text, add in methodological rigor for ethnographic approach, and frame the paper around the uncertainties of decision-making and how this is managed by staff in different positions. Also maybe more on role of family in this process.

Best of luck!

6. PLOS authors have the option to publish the peer review history of their article (what does this mean?). If published, this will include your full peer review and any attached files.

Reviewer #1: No

Reviewer #2: No

---

## [Author Response · Author response to Decision Letter 0]

2 May 2020

We would like to thank the reviewers very much for their positive remarks and constructive suggestions to improve the article.

Reviewer #1: 

Line 46: therefore not therefor 

Respond: Corrected line 46

Line 57: I would favor defining the ICU abbreviation in line 51 rather than line 57

Respond: Deleted line 57 and move to line 51

Please be sure that the manuscript adheres to the appropriate EQUATOR network guidelines and provide citation. Likely: O'Brien BC, Harris IB, Beckman TJ, Reed DA, Cook DA. Standards for reporting qualitative research: a synthesis of recommendations. Acad Med. 2014;89(9):1245-1251.

Respond: The manuscript is now following EQUATOR Guidelines and correct citations are made

Please state if the study was approved by an IRB. If so, state the organization whose IRB provided approval, and list the approval number in parentheses

Respond: The study was approved. Missing data added lines 282 - 285

Please state if informed consent was obtained and from whom. Was surrogate consent allowed?

Respond: Added lines 285-288

Please state if consent covered both study participation and publication of de-identified aggregate findings.

Respond: Described in lines 285 - 288

Line 182-183: Please update software reference: NVivo qualitative data analysis software (QSR International Pty Ltd, Doncaster, Australia)

Respond: Added in lines 221- 222

Line 199: defining NSIU should be done back up on line 58

Respond: Changed 

Please streamline the methods.

Please describe criteria for deciding when no further sampling was necessary (e.g., sampling saturation).

Please describe techniques to enhance trustworthiness and credibility of data analysis (e.g., member checking, audit trail, triangulation); rationale.

Respond: The section of methods has been streamlined (lines 123-167) and information of sampling saturation (lines 168-170) and how to enhance trustworthiness and credibility of data analysis has been described in lines 170- 178.

Reviewer #2:

The article addresses an important an important topic - underlying logics of early neurorehab - using an appropriate and under-utilised approach - ethnography. In general, the paper is clearly written, laid-out well, and provides relevant information to the reader. However, I have a number of queries which may need to be addressed in a re-framing of the paper. These are outlined below:

Respond: We do appreciate the general impression and the queries in need of been addressed in a re-framing and focusing paper. 

- The theoretical contribution of the paper is somewhat blurred. While the abstract states the paper investigates how concepts of time play out and help us understand underlying logics and dynamics of of social processes. There is also mention of 'temporal strategy games'. However the bulk of the paper focuses more broadly on the nature and objectives of early neurorehab rather than an in-depth look at the social dynamics or concepts of time. More about time in the macro sense (past, present, future) than micro temporal strategies of staff.

- Rather than concepts of time, I found the focus of the paper more on the slightly different - but also very interesting -uncertainty temporalities/alternative futures of clinical actions and decision-making in early neurorehab.

Respond: We are in the revised paper focusing on the temporality of uncertainty and decision making 

The changes in relation to this focus are now pervasive throughout the article. 

See: new title + lines 21-22, 72, 181, 216, 228, 233-238, 295- 303, 608-611, 636, 689-699. We have also added more data in the macro sense of strategies and conflicts of staff. Lines 358 - 383.

- Methods: Considering the ethnographic approach, I found the methods section lacking in a number of area:

Respond:This section has undergone major revision

o How were observations ‘participant’ as opposed just observation? [see Raymond Gold's typology]

Respond: Line 120: The word participant is deleted and the reference changed to Gold, R. (1958)

o How many hours observation?

Respond: Information added line 152

o Needs a reflective piece on impact of who did observations?

Respond: Information added, see line 124 + 142-150

o Was there any iteration between what you set out to observe and how data collection and analysis proceeded?

Respond: Information added 123-127

o How many ethnographic interviews were done and with who? There is a mentioned of transcribed interviews (p9) not sure how evident these are in findings

Respond: Information added line 151-158

Why focus group? How long? Discussion topics?

Respond: Section added line 158- 167

o Are relatives in interviews or observations? ‘Participated in three meetings with family members’ (p7) how?

Respond: Changed the words participated in to observed in order to clarify, line 138.

o Page 9 mentions hospital practice documents – not visible in findings

Respond: Deleted.

o Not always clear where data in findings is from (obs, interview, FG)

Respond: Added information, lines 348, 533, 626-627, 634, 671

o Integration of data not clear – focus group dialogue appears in abrupt manner

Respond: Clarified and changed

o Triangulation of data not evident (line 736)

Respond: Now described in lines 168- 178

- Linked to the above point, I have some concern about objectivity of analysis. At times analysis feels like it is selling NISU. See lines 244, 301-312, 400-403, 455-456, 616, 723-725, 749.

Respond: Thank you very much for this important point. We have looked into the mentioned lines and addressed the points as objectively as possible giving the situation that NISU is a single unit in the Danish healthcare system 

- Context section on NISU is far too long. Space of Opp section talks a lot about physical space but this doesn’t really feed into key findings – what does this section tell us, other than description of environment – which is then not taken up in discussion. No real discussion of time here either i.e. how time is entangled in physical space isn’t immediately obvious. Findings and discussion focus more on the decisions making and clinical action.

Respond: The context section is significantly shortened.

- Findings structured by; ‘space of opportunities, the clinical work, the assessment of potential’ which I didn't find very informative. An opportunity was missed here for deeper investigation of very important and theoretically rich topics which are only touched on. These include: decision-making rationalities, uncertain futures, how different disciplines negotiate uncertain futures – use of scoring/tools to navigate and manage uncertainty and ‘making the final call’ (564), responsibilities of final call between anaesthesiologists v other HCPs, embodied memories, active v passive rationalities of care (lines 460-463). All of these could be the main focus of the paper (rather than temporal strategies) - as they provide a richer account of the 'social processes' behind the practical clinical work.

Respond: We do find that the initial descriptions of NISU and the clinical practice must emerge in findings partly to allow the inexperienced reader with less knowledge of intensive work to gain a sufficient understanding. Furthermore, in terms of the fact that the NISU unit is singular in the Danish healthcare system, but also unusual in the international context, the descriptions are important.

However, we have added some sub-categories and added Figure 1 to give an overview of Findings and discussion.

- Must of the key findings and discussion are from the clinical work section – could focus here for re-framing of paper.

Respond: We see your points and agree. We have tried to reframe the paper accordingly

- While relatives are ‘crucial pieces’ (757), there is little emphasis on them in the findings.

Respond: This sentence has been removed from the conclusion 

- Discussion doesn’t follow through on potential contributions – how uncertainty is managed via situated/alternative futures. Overall, this paper could be much more concise in honing in on how staff manage the negotiation of futures which underpin early neurorehab clinical work.

Respond: Content order in the discussion section has been changed to match order and content in findings and the aim of the study.

- The link to theories in the literature are very broad, using concepts from mobile tech, history and political economy. There is little to no mention of a range of ethnographic healthcare (including intensive care) literature (see below). Could still use ‘anticipatory action’ concept but this need to be integrated more into medical/healthcare literature.

Respond: Thank you for enlightening us about relevant ethnographic healthcare literature. The literature is interesting, and one study has been included in the discussion section: Higginson IJ, Rumble C, Shipman C, Koffman J, Sleeman KE, Morgan M, Hopkins P, Noble J, Bernal W, Leonard S, Dampier O, Prentice.

Editor:

Respond: The manuscript is now following The PLOS ONE style templates.

2) In your Data Availability statement, you have not specified where the minimal data set underlying the results described in your manuscript can be found. PLOS defines a study's minimal data set as the underlying data used to reach theconclusions drawn in the manuscript and any additional data required to replicate the reported study findings in their entirety. All PLOS journals require that the minimal data set be made fully available. For more information about our data policy, please see http://journals.plos.org/plosone/s/data-availability.

Respond: The study’s minimal underlying data set (in Danish but if required, can be translated) is added as Supporting Information files. 

3) Thank you for your ethics statement : "The study was completed in accordance with the Helsinki Declaration of 2018. The regional committee for biomedical research ethics in Denmark had no objections to the study. https://doi.org/10.1080/09638288.2018.1501772. Informed written consent was obtained from the participating health professionals and the relatives of patients. All personal data were de-identified."

Please amend your current ethics statement to confirm if your named institutional review board or ethics committee specifically approved this study. Once you have amended this/these statement(s) in the Methods section of the manuscript, please add the same text to the “Ethics Statement” field of the submission form (via “Edit Submission”).

Respond: Ethical approval for this study was obtained by the Biomedical Research Ethics regional committee (1-16-02-91-17) and the Danish Data Protection Agency (journal no. 2012-58-006). The study was completed in accordance with the Helsinki Declaration 2008.

---

## [Decision Letter · Decision Letter 1]

18 May 2020

PONE-D-19-35410R1

The temporality of uncertainty in decision-making and treatment of severe brain injury. An ethnographic study of anticipatory action in early neurorehabilitation

PLOS ONE

Dear Mrs. Krogager Mathiasen,

Thank you for submitting your manuscript to PLOS ONE. After careful consideration, we feel that it has merit but does not fully meet PLOS ONE’s publication criteria as it currently stands. Therefore, we invite you to submit a revised version of the manuscript that addresses the points raised during the review process.

We would appreciate receiving your revised manuscript by Jul 02 2020 11:59PM. To enhance the reproducibility of your results, we recommend that if applicable you deposit your laboratory protocols in protocols.io, where a protocol can be assigned its own identifier (DOI) such that it can be cited independently in the future. For instructions see: http://journals.plos.org/plosone/s/submission-guidelines#loc-laboratory-protocols

We look forward to receiving your revised manuscript.

Kind regards,

Andrew Carl Miller

Academic Editor

PLOS ONE

Reviewers' comments:

Reviewer's Responses to Questions

**Comments to the Author**

1. If the authors have adequately addressed your comments raised in a previous round of review and you feel that this manuscript is now acceptable for publication, you may indicate that here to bypass the “Comments to the Author” section, enter your conflict of interest statement in the “Confidential to Editor” section, and submit your "Accept" recommendation.

Reviewer #1: All comments have been addressed

Reviewer #2: (No Response)

2. Is the manuscript technically sound, and do the data support the conclusions?

Reviewer #1: Partly

Reviewer #2: Yes

3. Has the statistical analysis been performed appropriately and rigorously? 

Reviewer #1: N/A

Reviewer #2: N/A

4. Have the authors made all data underlying the findings in their manuscript fully available?

Reviewer #1: Yes

Reviewer #2: No

5. Is the manuscript presented in an intelligible fashion and written in standard English?

Reviewer #1: Yes

Reviewer #2: Yes

6. Review Comments to the Author

Reviewer #1: - Rather than stating “first author”, consider defining Principle Investigator as PI and using that abbreviation thereafter.

- The importance of the science gets lost in all the storytelling and extraneous prose. This manuscript needs to be re-structured and significantly shortened to meet journal standards/format and to keep the reader engaged and “on task”.

Line 25: please be consistent with the explained definition of NISU that is used in the main text (neuro-intensive step-down unit)

- Line 26-28: Wording is a little confusing. Is the author intending to say: … negotiation of futures takes place in the modern ICU by strategically building upon past experiences.

- Line 31: present, not presence

- Line 31: consider rephrasing to … and desired futures.

- Introduction: In order to make the writing more concise, consider the following revision (not mandatory). I would advise starting the Introduction as follows (your choice): From the onset of a neurologic insult (e.g. traumatic brain injury, stroke, etc.) the patient’s experience of time and space changes [1, 42 2, 3]. Conversely, a clock begins from the health provider perspective, during which many believe an early window exists in which the brain's dynamic response to injury is heightened and rehabilitation is assumed to be particularly effective.

- Consider abbreviating severe acquired brain injury as this statement is used frequently.

- Line 103: has demonstrated

- Line 143: extent (not extend)

- The methods contain too much extraneous text and discussion for a medical journal. This needs to be tightened up and condensed. The theoretical framework (lines 179-217) should be condensed and included in the end of the introduction (before methods). Much of it could be removed.

- Line 142-150:I would recommend condensing this passage as follows: The primary investigator dressed in attire similar to other healthcare workers and wore a badge identifying the researcher role. The observer role proved non-disruptive to the teams, which are accustomed to those in observer roles including students and external providers.

- Line 153: conversations, not conservations

- Line 155: would it be more accurate to say surrogate, spouse, or family/friend, rather than wife? Specifying wife implies that all participants were married men (or lesbians). What if the person was un-married, or woman?

- Line 161: Was 1h 36min an average amount of time. It seems like an odd amount of time to specify a priori.

- Line 168-170: Sampling saturation occurred after 3 months of fieldwork.

- Line 170: ...credibility, the processes

- Line 170-172: ...generation were discussed and clarified during regular research team meetings, where PI analyses were validated by co-investigators.

- Line 173: endeavored

- Line 176: there are 2 sets of parentheses, so the sentence should end: ...theories)).

- Line 176-178: sentence fragment, please revise.

- Line 228-237: not necessary, would delete. This is still the methods. The findings are presented later.

- Context, Line 238-279: Should be condensed and moved up early in methds when discussing study environment and patient population.

- Line 284-285: The chosen verbage leaves the statement open to interpretation. I would favor just stating: Informed consent by a designated health care surrogate was permitted in cases where the patient did not have decision-making capacity.

- Line 292-294: no need to state what you are “going” to say in the results. Just state the results. I would delete these sentences.

- Results: Please start by stating the number of patients screened for study, the number consented, and the number included in the final analysis. Then state reasons for exclusion with patient numbers.

Reviewer #2: Revised edition is much improved. Methods much stronger. Appreciate that English may not be first language but there are some small grammatical errors throughout which may need to be picked up by editors. Have flagged as many as I could below:

Line 22: delete ‘is’

Line 30: ‘presence’ should be present

Line 111: delete particular

Line 112: observations

Line 122: intensive

Line 139: extent

Line 144: ‘stood aside’ to accompanied

Line 147-152: re: data collection – for clarity could state outright that analysis is based on 130 hours of observation, ?? formal (i.e. recorded) semi-structured interviews (if any or were they all informal, part of observational fieldwork? If so, state that they were informal, handwritten), and 1 focus group….

Line 165: ….generation were discussed…

Lines 167-171: is there a reference for use of data, investigator and theoretical triangulation?

Line 206: delete ‘for’

Line 211: interviews “and” observation

Line 215: also focus groups?

Line 257: remove bracket?

Line 284: is presented

Line 289: should be ‘main’ and ‘analysis’

Line 305: check patient

Line 349 nurses who expressed

Line 346-370: really interesting discussion from interviewee about advantages and disadvantages of NISU set-up. Almost seems like it might be better for patients but slightly worse for her in terms of her experience of professional autonomy. Could consider whether this inter-disciplinary impinging on autonomy experience would be better placed in clinical work or interdisciplinary sections?

Line 379: pleasant future life

Line 482: Inter-disciplinary Collaboration?

Line 509: delete ‘see next chapter’

Line 549: sentence may not be needed, previous sentences provide link to following section

Line 632-652: really interesting data – highlights how scores and navigational tools provide some sort of rationale (perhaps comfort for professionals) but the interdisplinary nature of decision-making means this is still a social process – shaped by other professionals’ opinions – this could be added to discussion (lines 805-808) re: ethical dilemmas of negotiating future lives.

Line 676-686: some repetition – intro of three logics not needed in both places. Suggest just at start of discussion.

Line 740-742: sentence meaning unclear

Line 812-814: not sure what this means – perhaps better to say that some of this practices in ICU and neuro-rehab are common and therefore findings may inform research in other settings?

I think the literature and discussion would benefit from the addition of one or 2 more socially focussed papers – as the findings are about the social dynamics and interprofessional collaboration of neuro-rehab e.g. Andreas Xyrichis, Karen Lowton, Anne Marie Rafferty (2017) Accomplishing professional jurisdiction in intensive care: An ethnographic study of three units - https://doi.org/10.1016/j.socscimed.2017.03.047

7. PLOS authors have the option to publish the peer review history of their article (what does this mean?). If published, this will include your full peer review and any attached files.

Reviewer #1: Yes: Andrew C. Miller

Reviewer #2: No

---

## [Author Response · Author response to Decision Letter 1]

23 Jun 2020

Reviewer #1: 

Rather than stating “first author”, consider defining Principle Investigator as PI and using that abbreviation thereafter.

- We have made the recommended change to P1

The importance of the science gets lost in all the storytelling and extraneous prose. This manuscript needs to be re-structured and significantly shortened to meet journal standards/format and to keep the reader engaged and “on task”. 

- The ethnographic descriptions have been shortened throughout the manuscript (approx. one page in total).

Line 25: please be consistent with the explained definition of NISU that is used in the main text (neuro-intensive step-down unit) 

- Changed 

Line 26-28: Wording is a little confusing. Is the author intending to say: … negotiation of futures takes place in the modern ICU by strategically building upon past experiences. 

- The sentence has been modified.

Line 31: present, not presence 

- Changed

Line 31: consider rephrasing to … and desired futures. 

- We have made the recommended change and deleted ‘certain hoped-for futures’

Introduction: In order to make the writing more concise, consider the following revision (not mandatory). I would advise starting the Introduction as follows (your choice): From the onset of a neurologic insult (e.g. traumatic brain injury, stroke, etc.) the patient’s experience of time and space changes [1, 42 2, 3]. Conversely, a clock begins from the health provider perspective, during which many believe an early window exists in which the brain's dynamic response to injury is heightened and rehabilitation is assumed to be particularly effective.

- Thank you. We have made the recommended change.

Consider abbreviating severe acquired brain injury as this statement is used frequently. 

- We have made the recommended change (sABI).

Line 103: has demonstrated 

- Changed

Line 143: extent (not extend) 

- Changed

The methods contain too much extraneous text and discussion for a medical journal. This needs to be tightened up and condensed. The theoretical framework (lines 179-217) should be condensed and included in the end of the introduction (before methods). Much of it could be removed.

- The section has been tightened up and condensed (deleted lines: 129-130, 172 175, 175 – 180, 120-122, 122- 130). We have chosen to keep the location of the theoretical framework. We find that the framework for the following analysis stands more clearly in this way. 

Line 142-150: I would recommend condensing this passage as follows: The primary investigator dressed in attire similar to other healthcare workers and wore a badge identifying the researcher role. The observer role proved non-disruptive to the teams, which are accustomed to those in observer roles including students and external providers. 

- We have made the recommended change.

Line 153: conversations, not conservations 

- Changed

Line 155: would it be more accurate to say surrogate, spouse, or family/friend, rather than wife? Specifying wife implies that all participants were married men (or lesbians). What if the person was un-married, or woman?

- Changed to “spouse”.

Line 161: Was 1h 36min an average amount of time. It seems like an odd amount of time to specify a priori.

- Changed to “about 1.5 hours”.

Line 168-170: Sampling saturation occurred after 3 months of fieldwork. 

- Modified, line 188-189.

Line 170: ...credibility, the processes 

- Changed

Line 170-172: ...generation were discussed and clarified during regular research team meetings, where PI analyses were validated by co-investigators. 

- Changed 

Line 173: endeavored 

- Changed

Line 176: there are 2 sets of parentheses, so the sentence should end: ...theories)). 

- Changed

Line 176-178: sentence fragment, please revise. 

- The sentence has been revised, line 196 line 

Line 228-237: not necessary, would delete. This is still the methods. The findings are presented later. 

- Deleted

Context, Line 238-279: Should be condensed and moved up early in methods when discussing study environment and patient population. 

- Condensed. The recommend lines have been deleted and moved up as recommended to line 134.

Line 284-285: The chosen verbage leaves the statement open to interpretation. I would favor just stating: Informed consent by a designated health care surrogate was permitted in cases where the patient did not have decision-making capacity. 

- Changed 

Line 292-294: no need to state what you are “going” to say in the results. Just state the results. I would delete these sentences. 

- The sentences are deleted.

Results: Please start by stating the number of patients screened for study, the number consented, and the number included in the final analysis. Then state reasons for exclusion with patient numbers. 

- Information has been added in the section “Data collection”, line 181-187.

Reviewer #2: 

Revised edition is much improved. Methods much stronger. Appreciate that English may not be first language but there are some small grammatical errors throughout which may need to be picked up by editors. Have flagged as many as I could below: 

- Thank you for your comment about the revised edition and for your help with the grammatical errors. 

Line 22: delete ‘is’ 

- Error is corrected 

Line 30: ‘presence’ should be present 

- Error is corrected

Line 111: delete particular 

- Changed

Line 112: observations 

- Error is corrected

Line 122: intensive 

- Error is corrected

Line 139: extent

- Error is corrected

Line 144: ‘stood aside’ to accompanied 

- Sentence has been changed 

Line 147-152: re: data collection – for clarity could state outright that analysis is based on 130 hours of observation, ?? formal (i.e. recorded) semi-structured interviews (if any or were they all informal, part of observational fieldwork? If so, state that they were informal, handwritten), and 1 focus group…. 

- The sentences have been clarified 

Line 165: ….generation were discussed… 

- Changed

Lines 167-171: is there a reference for use of data, investigator and theoretical triangulation? 

- Reference added, line 197

Line 206: delete ‘for’ 

- Deleted

Line 211: interviews “and” observation 

- Error is corrected

Line 215: also focus groups? 

- Added

Line 257: remove bracket? 

- Removed

Line 284: is presented 

- The sentence is been deleted

Line 289: should be ‘main’ and ‘analysis’ 

- Changed, line 264

Line 305: check patient 

- Changed

Line 349 nurses who expressed 

- Changed, line 432

Line 346-370: really interesting discussion from interviewee about advantages and disadvantages of NISU set-up. Almost seems like it might be better for patients but slightly worse for her in terms of her experience of professional autonomy. Could consider whether this inter-disciplinary impinging on autonomy experience would be better placed in clinical work or interdisciplinary sections? 

- The discussion has been moved to The interdisciplinary section, line 429 - 454.

Line 379: pleasant future life 

- Changed, line 323

Line 482: Inter-disciplinary Collaboration? 

- Changed to The clinical interdisciplinary collaboration, line 409

Line 509: delete ‘see next chapter’ 

- Deleted

Line 549: sentence may not be needed, previous sentences provide link to following section. 

- The sentence is been deleted.

Line 632-652: really interesting data – highlights how scores and navigational tools provide some sort of rationale (perhaps comfort for professionals) but the interdisplinary nature of decision-making means this is still a social process – shaped by other professionals’ opinions – this could be added to discussion (lines 805-808) re: ethical dilemmas of negotiating future lives. 

- Information added in line 755 – 759.

Line 676-686: some repetition – intro of three logics not needed in both places. Suggest just at start of discussion. 

- Deleted

Line 740-742: sentence meaning unclear 

- We have not been able to locate the sentence.

Line 812-814: not sure what this means – perhaps better to say that some of this practices in ICU and neuro-rehab are common and therefore findings may inform research in other settings? 

- Deleted 

I think the literature and discussion would benefit from the addition of one or 2 more socially focused papers – as the findings are about the social dynamics and interprofessional collaboration of neuro-rehab e.g. Andreas Xyrichis, Karen Lowton, Anne Marie Rafferty (2017) Accomplishing professional jurisdiction in intensive care: An ethnographic study of three units - https://doi.org/10.1016/j.socscimed.2017.03.047

- Addition added line 760-768

---

## [Editor Report · Decision Letter 2]

19 Aug 2020

The temporality of uncertainty in decision-making and treatment of severe brain injury. An ethnographic study of anticipatory action in early neurorehabilitation

PONE-D-19-35410R2

Dear Mia Krogager Mathiasen and Hanne Pallesen,

We’re pleased to inform you that your manuscript has been judged scientifically suitable for publication and will be formally accepted for publication once it meets all outstanding technical requirements.

Kind regards,

Vardan Karamyan, Pharm.D., Ph.D.

Academic Editor

PLOS ONE
---

## [Editor Report · Acceptance letter]

21 Sep 2020

PONE-D-19-35410R2 

The temporality of uncertainty in decision-making and treatment of severe brain injury 

Dear Dr. Krogager Mathiasen:

I'm pleased to inform you that your manuscript has been deemed suitable for publication in PLOS ONE. Congratulations! Your manuscript is now with our production department. 

Kind regards, 

on behalf of

Dr. Vardan Karamyan 

Academic Editor

PLOS ONE